# Study on Mechanical Properties and High-Speed Impact Resistance of Carbon Nanofibers/Polyurethane Composites Modified by Polydopamine

**DOI:** 10.3390/polym14194177

**Published:** 2022-10-05

**Authors:** Feng Qi, Jun Gao, Bolun Wu, Hongyan Yang, Fugang Qi, Nie Zhao, Biao Zhang, Xiaoping Ouyang

**Affiliations:** 1School of Materials Science and Engineering, Xiangtan University, Xiangtan 411105, China; 2Qingdao Green World New Material Technology, Qingdao 266100, China; 3Key Laboratory of Low Dimensional Materials and Application Technology of Ministry of Education, Xiangtan University, Xiangtan 411105, China

**Keywords:** polyurethane elastomers, dynamic impact, CNFs, energy absorption capacity

## Abstract

Polyurethane elastomers (PUE), with superior mechanical properties and excellent corrosion resistance, are applied widely to the protective capability of structures under low-speed impact. However, they are prone to instantaneous phase transition, irreversible deformation and rupture even arising from holes under high-speed impact. In this paper, mussel adhesion proteins were applied to modify carbon nanofibers (CNFs) in a non-covalent way, and creatively mixed with PUE. This can improve the dispersity and interfacial compatibility of nanofillers in the PUE matrix. In addition, the homogeneous dispersion of modified nanofillers can serve as “reinforcing steel bars”. The nanofillers and PUE matrix can form “mud and brick” structures, which show superb mechanical properties and impact resistance. Specifically, the reinforcement of 1.0 wt.% modified fillers in PUE is 103.51%, 95.12% and 119.85% higher than the neat PUE in compression modulus, storage modulus and energy absorption capability, respectively. The results have great implications in the design of composite parts for aerospace and army vehicles under extreme circumstances.

## 1. Introduction

Polyurethane elastomers (PUEs) are a polymer material with a flexibly alternating arrangement of soft segments (SS) and hard segments (HS) to satisfy various needs of engineering applications, especially in dynamic events [1,2], such as protective enclosures and human body protection, against low-speed impact, on account of their excellent impact-resistant performance, such as shearing stiffening and strengthening, shock absorption and fatigue resistance [1,3,4,5]. However, their irreversible deformation, instantaneous phase transition and rupture failures under high-speed impact limit their applications in armor and vehicle protection [1,5,6].

High-speed impact occurs when high-velocity and high-energy shot/debris impact the target, which means that flexible protective equipment provides superior strength and resilience to absorb impact energy and avoid penetration [7,8]. According to research in recent years, researchers have designed many strategies to improve the impact resistance of PUE composites via changing reactant polyurethane [9,10], forming the interpenetrating structure [11,12], or adding organic/inorganic fillers [1,4,5,13,14,15,16]. Therein, inorganic nanofillers of the studies are more extensive and practical, and have become one of the major effect ascensions of PUE performance [1,5]. As nanoscale fillers, most of them have no defects, and their applications in the field of polymer composites have developed a new prospect to overcome the limitation of traditional micron and larger-scale fillers [17,18]. Therefore, carbon nanofibers (CNFs), which have superior strength, thermostability and electroconductibility, are cost-effective materials that have the potential to replace carbon nanotubes [19,20]. However, the surface of nano-carbon fibers are short of active groups, showing chemical inertia, and they are easy to agglomerate due to high aspect ratio and van der Waals attraction of the nanofillers [21,22,23,24], thus their poor dispersion in polymer matrices are often referred to as the sources of cracks, hindering their wide application [18,22]. Therefore, it is necessary to modify CNFs to make them disperse evenly in the matrix and make full use of the enhancement of nanofillers. In addition, effectively designing interfacial interactions between CNFs and blending components can achieve controlled microstructure evolution and ultimate properties [1,8,25]. Multiple surface functionalization uses for nanofillers include covalent and non-covalent approaches; covalent approaches can damage the local plane and rigid framework of nanofillers, which leads to a reduction in the mechanical properties of the composites, while non-covalent approaches do not damage the mechanical properties of nanofillers [26,27,28,29,30]. Chen et al. [29] used PDA to non-covalently modify h-BN via improving the dispersity and interfacial compatibility in the benzoxazine polymer and improved the thermal conductivity and thermal stability of polymer composites without damaging their mechanical properties. Xiao et al. [30] used 1-pyrenecarboxylic acid to non-covalently modify the CNT and improved electrical and tensile properties of the modified thermoplastic polyurethane composites due to more uniform CNT dispersion in the matrix. While covalent modification has been considered to be an effective method to enhance the interfacial interaction between the surface modified fillers and the polymer matrix, the surface of inert nanofiller is damaged inevitably by strong oxidizing agent or acid [28].

Natural mussels can robustly adhere to ships or rocks in turbulent waves, and some studies have found that mussel adhesive proteins, which contain covalent bonding and non-covalent bonding to improve the interfacial binding between substrate and adhesive proteins, have strong adhesion ability and can adhere to almost every substrate [11,31,32]. Additionally, related studies found that the secret of adhesion proteins is from mussel foot-byssal proteins (Mfps), in which the catechol and amido play an essential role in adhesive behaviors [11,33,34,35,36]. In the past ten years, many polymer-reinforcing fillers have been modified by polydopamine (PDA) inspired by the mussel adhesive effect, which is deemed to an effective way to functionalize reinforcing fillers and make them compatible in the polymer matrix [37,38,39,40]. However, there are very few studies on the dynamic mechanical properties of polyurea or polyurethane nanocomposites under high-speed impact.

PUEs with excellent characteristics have a potential in high-speed impact applications after specific treatment. The previous research methods are complicated in process and high in cost, which are not suitable for mass production. Our experimental process is simple and suitable for large-scale application; the practical effect is remarkable and the cost is low. Inspired by mussel adhesive proteins in mussels, we used commercial dopamine hydrochloride to bionic design mussel adhesion proteins so as to improve the interfacial bonding between CNFs and PUEs, as shown in Figure 1, and studied the dynamic properties and solid viscoelastic properties of PUE composites. The experimental data indicated that the addition of only tiny PDA@CNFs could significantly increase the mechanical property and dynamic impact resistance of PUE composites.

## 2. Material and Methods

### 2.1. Materials

The PUEs, which consisted of an A component (2,4-TDI, –NCO, 6.212 wt.%) and B component (poly(oxycarbonyloxy-1,6-hexanediyl), –OH, 6.358 wt.%), were provided by Qingdao Green World New Material Technology Co., Ltd. Carbon nanofibers (CNFs, ≥97%, diameter: 50–200 nm, length: 1–15 μm) were purchased from Xianfeng Nanotechnology Co. Other chemicals, including dopamine hydrochloride (DA, ≥98.0%), hydrochloric acid (HCl, AR, 36.0–38.0%) and Tris-HCl (AR, ≥99.0%) were purchased from Aladdin.

### 2.2. Preparation of PDA @CNFs

A total of 0.1 g Tris-HCl and 200 mL H_2_O were added into the container, 0.05 mol/L HCl was used to adjust the PH to 8.5. 0.2 g DA, and 0.7 g CNFs were added into a flask and stirred at 25 °C for 24 h. They were washed several times with water to neutral, then filtered and dried at 60 °C for 24 h. These modified CNFs were named PDA@CNFs–1. Keeping the other reaction conditions constant, the concentration of DA was changed to 0.4 g and 0.8 g, named as PDA@CNFs–2 and PDA@CNFs–3, respectively. Table 1 shows the detailed information of the nanomaterials.

### 2.3. Preparation of PDA@CNFs/PUE (PC/PUE)

The PUEs and PDA@CNFs were added to the beaker and stirred for 10 min, and were put into a vacuum deaeration machine to pump air bubbles of PUE composites for 5 min. Finally, they were poured into the mold of the specified size and cured for 5 days. Thereinto, the mass ratio of A component to B component was maintained at 5:2. We controlled the mass fraction of these three fillers at 1.0 wt.%, and then successively added PUEs to prepare samples, named as PDA@CNFs/PUE–1, PDA@CNFs/PUE–2 and PDA@CNFs/PUE–-3. It was considered that PDA@CNFs–1 had the best dispersion in ethanol as shown in Figure 2, and from the subsequent static compression experiments, it was found that PDA@CNFs/PUE–1 had the best mechanical properties. Therefore, we continued to study the effect of mass fraction of PDA@CNFs–1 on PUE composites, and the mass fraction of fillers was 0.2%, 0.5%, 0.8%, 1.0% and 1.2% of the mass of PUE matrix, named as PC/PUE–Ⅰ, PC/PUE–Ⅱ, PC/PUE–Ⅲ, PC/PUE–Ⅳ and PC/PUE-Ⅴ, respectively. The 1.0 wt.% CNFs were mixed with PUE and served as the control groups, named as CNFs/PUE. Table 2 shows the information of PUE composites in detail.

### 2.4. Characterizations

The FTIR (Thermo Scientific, Nicolet iS5, Waltham, USA) was obtained using a Nicolet 6700 with a scanning range of 3950–525 cm^−1^. The polymerization of CNFs was detected by a thermal gravimetric analyzer (TGA, TGA5500, New Castle, DE, USA) in N_2_ at a heating rate of 20 °C/min. The surface chemical components of the nanoparticles were characterized by X-ray photoelectron spectroscopy (XPS, Thermo Scientific K-Alpha, Palo Alto, CA, USA). Absorption of the aqueous solution of nanofillers was measured with UV-VIS (Agilent, CARY100, New Castle, DE, USA) from 550 nm to 260 nm. Scanning electron microscopy (SEM, Zeiss, Sigma300, Baden-Württemberg, Germany) was used to observe the distribution of nanofillers and fracture surface morphology of PUE composites after the SHPB test. An optical microscope (Olympus, BX53M, Tokyo, Japan) was used to observe the dispersity of CNFs and PDA@CNFs and surface damage conditions of PUE composites after dynamic impact. The universal electronic testing machine (Hua Long, WDW-100C, Changsha, China) was used for static compression test of PUE composites (Simples: Φ20*4 mm, cylinder). Compression test parameters (such as compression modulus and compression resilience rate) were referenced from ISO 7743 2008(E) and GB/T 20671.2-2006, as shown in Formulas (1) and (2). The compression modulus was used to characterize the ability of the sample to resist deformation. The compression resilience rate was used to characterize the resilience of the material after compression.
Compression modulus (MPa) = F/A(1)F: Compressive force (N)A: Initial cross-sectional area of sample (mm^2^)


Compression resilience rate (%) = [(R-M)/(P-M)] ∗ 100(2)
P: Sample thickness under initial loadM: Sample thickness at full loadR: Thickness of sample rebound


The strain rate of tests was set at 0.005 s^−1^. Three parallel experiments were conducted for all samples to take the average value. Dynamic mechanical analysis (DMA, Eplexor500N, Würzburg, Germany) was used to study the dynamic mechanical properties and interface problems of the solid viscoelastic properties. DMA test conditions were as follows: sample size: cuboid, 35 × 5 × 2 mm; tensile strain: 1.0%; dynamic strain: 0.2%; heating rate: 5 °C/min; frequency: 5 Hz; test temperature range: –60°C–70 °C. The dynamic impact resistance of composite PUEs (samples: Φ10*2 mm, cylinder) were measured by a split Hopkinson pressure bar (SHPB, National University of Defense Technology, Changsha, China) made of aluminum alloy.

## 3. Results and Discussion

### 3.1. Structure Characterizations

An optical microscope was used to observe the dispersity of nanoparticles in an ethanol solution after 10 min ultrasonic and setting for 10 days, as shown in Figure 2. In contrast to the agglomeration of CNFs (see Figure 2a), PDA@CNFs had a better dispersity property in an ethanol solution (see Figure 2b–d); in particular, PDA@CNFs-1 were best. FT-IR, UV/Vis, XPS and TGA were used to prove the successful synthesis of PDA@CNFs. From the FT-IR spectra in Figure 3a, new absorption peaks of N-H (1506 cm^−1^), C-N (1231 cm^−1^) and C-O (1110 cm^−1^) are seen. As shown in Figure 3b, a new signal, N1s at 400.1 eV, appears in the spectra of PDA@CNFs, and N1s accounts for 3.61% of PDA@CNFs. Figure 3c shows the UV-VIS spectra chemical polymerization of DA after 24 h at 25 °C, and PDA@CNFs shows two broad peaks at around 280 nm and 405 nm in the aqueous solution, which are suggested as evidence for the oxidation self-polymerization of DA [41,42]. For CNFs, the XPS C1s spectrum can be curve-fitted with four peak components at the binding energy of 284.8 eV, 286.6 eV, 289.0 eV and 291.1 eV, attributable to the C-C, C-O, O=C-O and π-π species, respectively, as shown in Figure 3d. However, for PDA@CNFs, the XPS C1s spectra shows a new peak at 285.5 eV, which represents C-N (see Figure 3e). From Figure 3f, we find that CNFs lose about 5.2% weight at 800 °C, due to evaporation of water and oxide decomposition. As for the different concentrations of PDA to modify CNFs, the weight of PDA@CNFs decreases with the temperature arising, and the weight of fillers containing a high concentration of PDA decreases more. TGA spectra of PDA@CNFs can be divided into three stages. Taking PDA@CNFs–1, for instance: at 30–150 °C, nanofillers lose almost 2% of weight on account of absorbed water; at 150~580 °C, nanofillers lose about 7% of weight due to decomposition of PDA; at 580–800 °C, nanofillers lose about 2% of weight because of the decomposition of oxide on CNFs. The above experiments can prove that we succeed in synthesizing PDA@CNFs.

The microtopography of nanofillers can be observed by SEM and TEM. Figure 4 shows the surface morphology of CNFs and PDA@CNFs–1. There are 2 nm holes in the CNFs (see Figure 4a_3_). Compared with the smooth surface of CNFs (see Figure 4a), the surface of PDA@CNFs–1 (see Figure 4b) is rougher and thicker, and is wrapped by PDA. The thickness of coating on the surface of PDA@CNFs–1 is about 5 nm.

### 3.2. Mechanical Properties of PUE Composites

#### 3.2.1. Static Compression Analysis

Considering that PUEs exhibit obvious viscoelasticity and viscoplasticity during the compression process, it is necessary to study the compression modules and the compression resilience rate to indicate their strength and toughness [1,43]. It can be seen from the initial stage that the stress–strain curve is linear, and the corresponding slope corresponds to the compression modulus, also known as the elastic modulus (see Figure 5a_1_). Figure 5 shows the stress–strain curve, compression modulus and compression resilience rate after compression test of PUE composites. There is no obvious deformation and damage of PCs/PUEs after experiment, while CNFs/PUEs and the neat PUEs display serious irreversible deformation and failure.

From Figure 5a_1_,b_1_, the strain–stress curves have excellent repeatability. Leaving the mass fraction of nanofiller as 1.0 wt.%, we find that PDA@CNFs/PUE–1 have better mechanical properties: higher compression modulus, compression resilience rate and maximum compressive strength. Besides, it is worthwhile to study the effect of different mass fractions of nanofillers on PUE. Therefore, we continued to design the five different mass fractions of PDA@CNFs–1 due to their good dispersity. The data show that, with the mass fraction increase of PDA@CNFs–1, the compression modulus, compression resilience rate and maximum compressive strength of composite PUEs first rise then fall, as shown in Table 3 and Figure 5. Among them, PC/PUE–Ⅳ has excellent mechanical properties: compression resilience rate is 82.6%, compression modulus is increased by 103.54%, and the maximum compressive strength is enhanced by 141.25% compared with the neat PUE. These excellent properties are attributed to the dispersity and compatibility between PDA@CNFs and PUEs, while the PUE with CNFs increased the strength alone, and the compression resilience rate was quite low. However, the mechanical performance of PC/PUE-Ⅴ was degraded because superabundant PDA@CNFs hinder the internal movement and friction of PUE segments.

The addition of PDA@CNFs can significantly influence the strength and toughness of PUE composites during the static compression test, which are beneficial for the following dynamic experiments.

#### 3.2.2. Dynamic Impact Performance

A smaller aspect ratio (L/D = 0.2) is favorable to avoid the inertia effect and end effect, and to maintain uniform deformation and stress balance more quickly, especially for soft materials such as PUEs [1,5,43]. The composition and parameter data of the SHPB apparatus schematic diagram is shown in Figure 6a. The apparatus consists of four bars. All four bars are made of Al alloy (elastic modulus E = 70 GPa, Poisson’s ratio υ = 0.33, density ρ = 2.8 g/cm^3^, diameter Φ = 20 mm). Because the PUEs exhibited strain-rate dependency of its mechanical properties, it was necessary to control the same strain rate by controlling the gas pressure parameters of the apparatus at 20 °C to measure the dynamic impact effect on different mass fraction nanofillers on PUEs [1,43,44]. After adjusting the instrument, the strain–time curve of the interface between the incident bar and the sample (incident wave + reflected wave) coincided with that between the transmitted bar and the sample (transmitted wave), indicating that the sample achieved a good stress balance during the loading process [45,46], as shown in Figure 6b. The results show that yielding was not visible and strain rate was controlled at about 9000 s^−1^ during the SHPB test (see Figure 6c). The experimental graphs after the SHPB test are as in Figure 7, where the neat PUEs and CNFs/PUEs exhibit impact perforation failure, while the PC/PUE-Ⅳ exhibits tiny radial and axial crazes without obvious cracks or deformation (see Figure 7d,e). The stress–strain curve and strain energy density for all kinds of PUE composites are shown in Figure 8. The strain energy density of PC/PUE, which is used to reflect the impact-resistance capacity of the specimens, is defined as the area under the stress–strain curve [1,47]. We find that the addition of PDA@CNFs affects the dynamic strength, elongation and absorbed energy capacity of the specimens. For example, when the addition of PDA@CNFs reaches 1.0 wt.%, the maximum dynamic strength of PC/PUE–Ⅳ reaches up to 361.93 MPa, which is enhanced about 130.36% compared with the neat PUE. The elongation of PDA@CNFs-Ⅳ reaches 1.25, higher than CNFs/PUE, and strain energy density is improved by 78.86% compared with the neat PUE. These results indicate that PDA@CNFs/PUEs can play a great role of preventing crack growth by the bridging effect or inducing the crack deflection. The excellent impact resistance of the specimens is attributed to the dispersity of PDA@CNFs and the compatibility between PDA@CNFs and PUEs. Besides, the PDA on the CNFs can offer covalent bonds and noncovalent bonds, such as hydrogen bonds, π-π conjugation, electrostatic adsorption, and –NH_2_/–NH, –OH, crosslinking with PUEs, which can form flexible and strong connection. The PUE composites with such special characteristics are able to withstand permanent damage and failure under high-speed impact load, which reveals their potential applications for military and personal protection engineering.

#### 3.2.3. Damage Morphology Analysis

The mechanical enhancement of polymer nanocomposites (PNCs) system mainly depends on the dispersion and interfacial compatibility of nanofillers in the matrix. Figure 9 is the axial SEM images of the samples after SHPB test, which are cooled in liquid nitrogen. Figure 9a display the neat PUE fracture surface with obvious perforation and many cracks near the impact hole, but relatively smooth away from the hole, due to stress concentration of high-speed impact. As is shown in Figure 9b, compared with the neat PUE, CNFs/PUE present many wrinkles and a large amount of CNFs agglomeration behaviors in PUE matrix, which can hinder the propagation direction of the impact crack to some extent and enhance the impact resistance of materials. However, the agglomeration behaviors of CNFs can be the sources of impact failure. PC/PUE–Ⅳ have many wrinkles and the nanofillers have a good dispersity and interfacial compatibility. PDA@CNFs are served as “reinforcing steel bars” relatively uniform arrangement in the PUE matrix to fixed the PUE and prevent cracks propagation by bridging effect or inducing the crack deflection, which are beneficial to stress transmission and energy absorption. The “reinforcing steel bars” and soft PUE can form “mud and brick” structures, which enhance the impact resistance of PUE composites. In addition, the plastic deformation and dimple formation of the joints are mainly related to the orientation and flow of the joints and the volume expansion of the joints under loading conditions, which consumes a lot of energy.

### 3.3. Mechanism of Interfacial Strengthening of PUE Composites

Under extreme dynamic loading conditions, PUEs undergo a transient phase transition. When the pulse velocity approaches the corresponding segmental mobility, PUEs change from rubber to leather or even glass, which greatly increases the stress level and realizes greater energy dissipation [5].

DMA tests have been widely applied to characterize the interface properties and solid viscoplasticity of heterogeneous PNCs. The variations in dissipation factor (Tan δ) and storage modulus (E’) with temperature are shown in Figure 10. T_g_ is usually taken as the temperature corresponding to the peak of tan δ, reflecting the fluidity of polymer segments during experimental process [48].

The T_g_ of neat PUE is about –35.6 °C. It is obvious from the results that with an increase of content of PDA@CNFs, the E’ values soared dramatically while T_g_ first fell and then rose accordingly during the whole temperature range, as shown in Figure 10. In comparison to the neat PUE, CNFs/PUEs exhibited higher E’ and higher T_g_, which shows that CNFs can endow PUEs in strength but not toughness. However, PDA@CNFs have a potential to reduce the T_g_ of PUE composites with a small amount of mass fraction. In the cryogenic glass zone above the glass transition zone, the value of E’ increases significantly, because of elasticity and chain migration confinement effect of PDA@CNFs in the PUE matrix, while the Tan δ peak heights reduce and shift toward higher temperatures, which shows that adding more modified fillers will cause partial agglomeration of polymer composites to restrict the movement of chain joints and cause a decrease in the toughness of the material. Experiments verified that the modification and dispersion of CNFs in the PUE matrix can cause hydrodynamic enhancement effects, and the dispersion of nanofillers in the PUE is controlled by the modified state and mass fraction of nanofillers.

## 4. Conclusions

Our works focus on the dynamic thermomechanical response and high-speed impact performance of PUE composites modified by PDA, and the effects of PDA@CNFs on PUE are discussed from the perspectives of chemical composition and structures. PDA can serve as “interfacial agents”, which provide π-π conjugation and hydrogen bonds, noncovalent bonding, and robust covalent bonding, such as amidogens and hydroxyls, crosslinking PUE to improve the interface interaction between CNFs and PUEs. We studied the effect of three different concentrations of DA to modify CNFs in PUE composites, and continued to investigate the effect of PDA@CNFs–1 mass fraction of PUE composites. To systemically study the mechanical properties of PDA@CNFs/PUE, we carried out static compression tests, SHPB tests and DMA experiments, then we observed the sections of PUE composites after SHPB tests. The experimental data show that adding tiny PDA@CNFs can significantly endow PUEs with a comprehensive mechanical performance and high-speed impact resistance via improving homogeneous dispersion, promoting interfacial bonding. The main experimental conclusions are as follows.

(1)PDA can improve the dispersity of CNFs and compatibility between CNFs and PUE matrices.(2)PDA@CNFs/PUEs have excellent and comprehensive mechanical properties in static compression tests. The compression modulus and maximum strength of PC/PUE-Ⅳ were increased by 103.54% and 141.25%, respectively, compared with the neat PUE. The compression resilience rate of PC/PUE–Ⅳ was up to 82.6%, which was much better than the neat PUE and CNFs/PUE.(3)During the dynamic impact tests and DMA tests, PC/PUEs had excellent energy absorption capability, storage modulus and impact penetration resistance due to good dispersity and interface combination and “mud and brick” structures in the PUE composites.

## Figures and Tables

**Figure 1 polymers-14-04177-f001:**
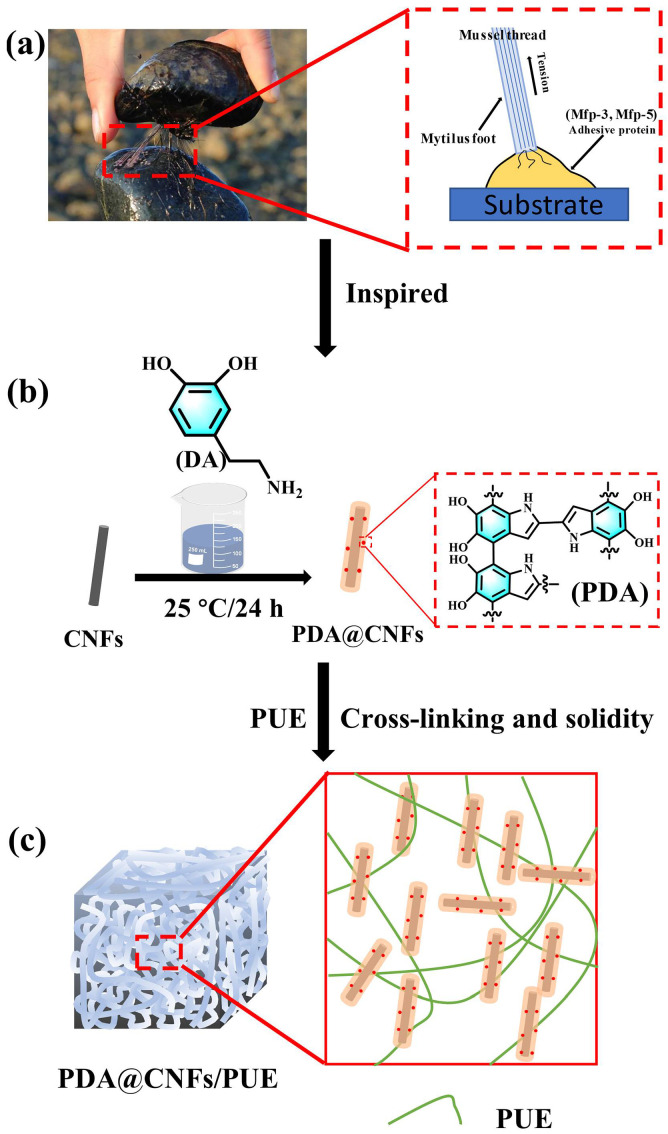
(**a**) Mechanism of mussel adhesion proteins, (**b**) synthesis of PDA@CNFs, (**c**) schematic diagram of PDA@CNFs/PUE covalent and non-covalent conjunction with PUEs.

**Figure 2 polymers-14-04177-f002:**
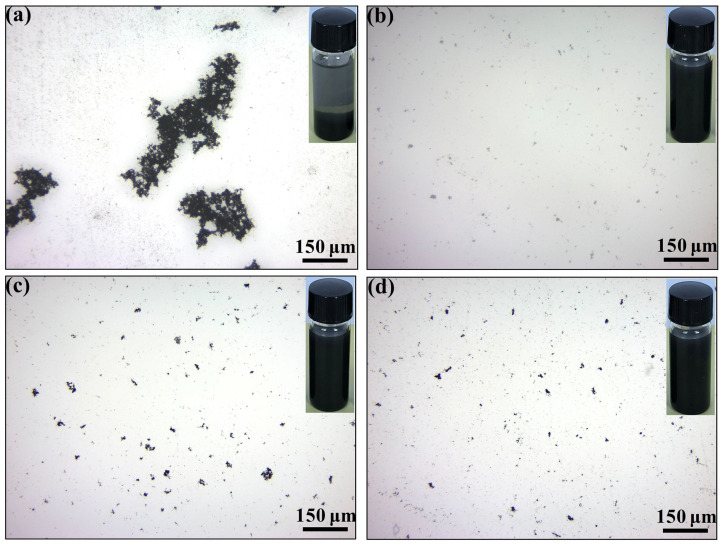
Optical microscope and macroscopic image of nanofillers dispersing in ethanol solution after 10 min ultrasonic and setting for 10 days. (**a**) CNFs, (**b**) PDA@CNFs–1, (**c**) PDA@CNFs–2, (**d**) PDA@CNFs–3.

**Figure 3 polymers-14-04177-f003:**
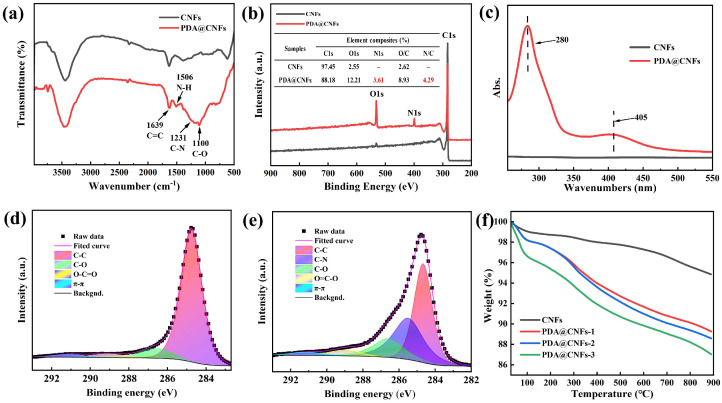
(**a**) FT-IR spectra of CNFs and PDA@CNFs, (**b**) XPS full scanning spectra and the element content of the nanofillers, (**c**) UV-VIS spectra of PDA@CNFs and CNFs, C1s spectra of (**d**) CNFs and (**e**) PDA@CNFs, (**f**) TGA spectra of CNFs and different concentration gradients of PDA to modified CNFs.

**Figure 4 polymers-14-04177-f004:**
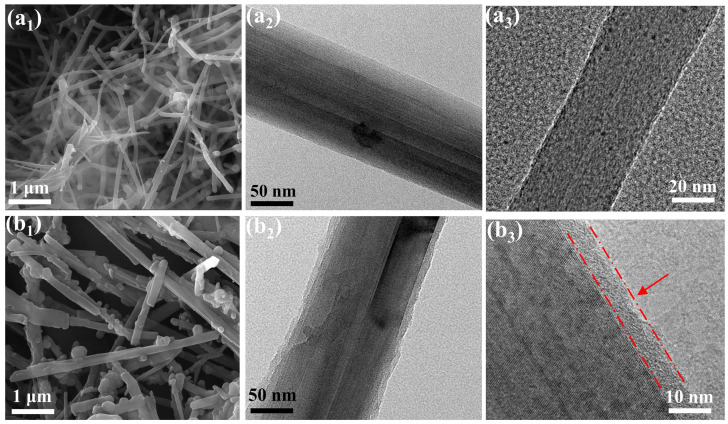
SEM (**a_1_**,**b_1_**) and TEM (**a_2_**,**a_3_**,**b_2_**,**b_3_**) images of (**a**) CNFs, (**b**) PDA@CNFs–1.

**Figure 5 polymers-14-04177-f005:**
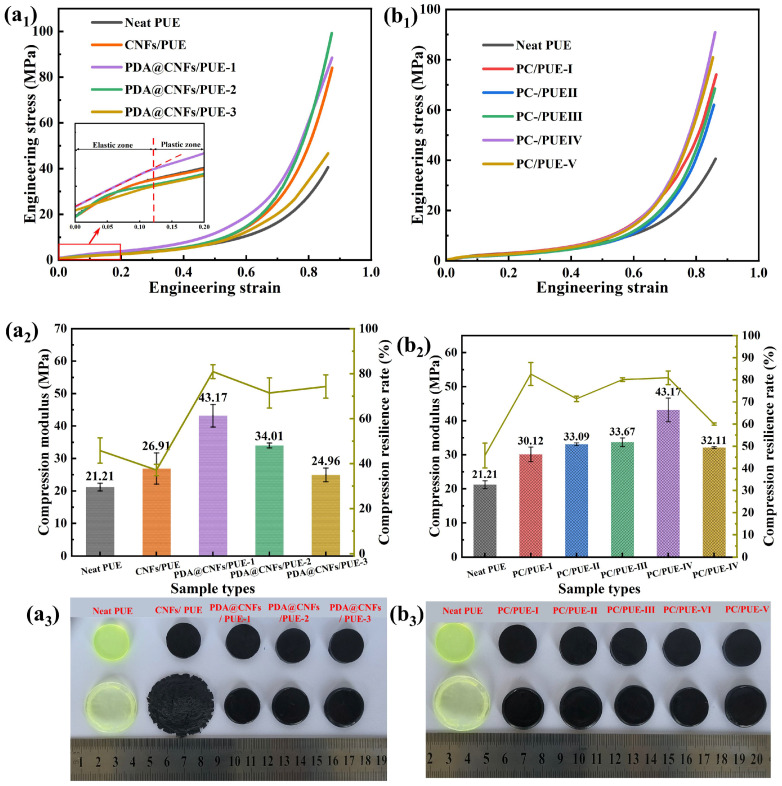
(**a_1_**) Stress–strain curves of difference concentration of DA to modify CNFs in PUE matrix. (**a_2_**) Compression modulus and compression resilience rate of difference concentration of DA to modify CNFs in PUE matrix. (**a_3_**) Experimental graphs of PUE composites after static compression tests of difference concentration of DA to modify CNFs in PUE matrix. (**b_1_**) Stress–strain curves of different mass fraction at concentration of PDA@CNFs–1 in PUE matrix. (**b_2_**) Compression modulus and compression resilience rate of different mass fraction at concentration of PDA@CNFs–1 in PUE matrix. (**b_3_**) Experimental graphs of PUE composites after static compression tests of different mass fraction at concentration of PDA@CNFs–1 in PUE matrix.

**Figure 6 polymers-14-04177-f006:**
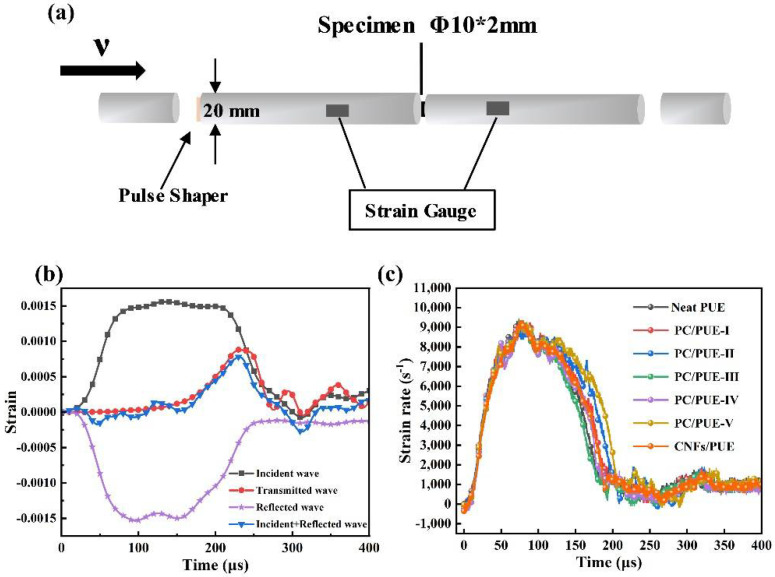
(**a**) SHPB apparatus schematic diagram, (**b**) raw data for SHPB calibration, (**c**) the strain rate of PUE composites after SHPB impact.

**Figure 7 polymers-14-04177-f007:**
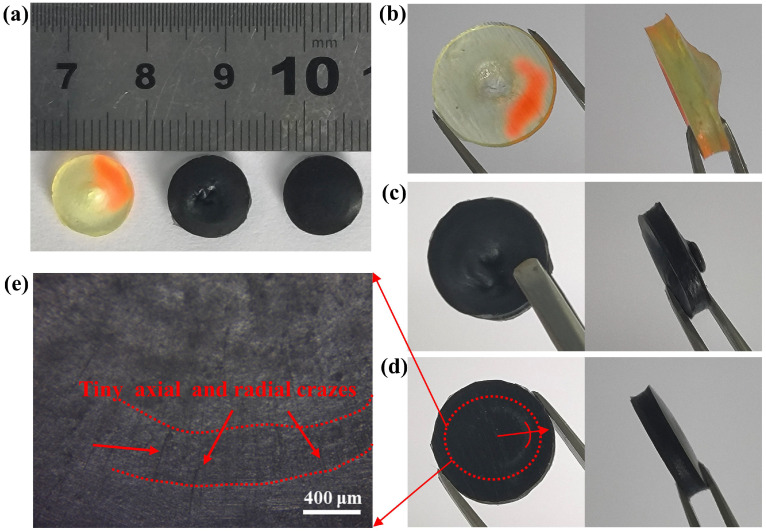
(**a**) Pictures of neat PUE, CNFs/PUE and PC/PUE-Ⅳ after SHPB test (from left to right), magnified pictures of (**b**) the neat PUE, (**c**) CNFs/PUE, (**d**) PC/PUE–Ⅳ. (**e**) Optical microscope image of PC/PUE–Ⅵ.

**Figure 8 polymers-14-04177-f008:**
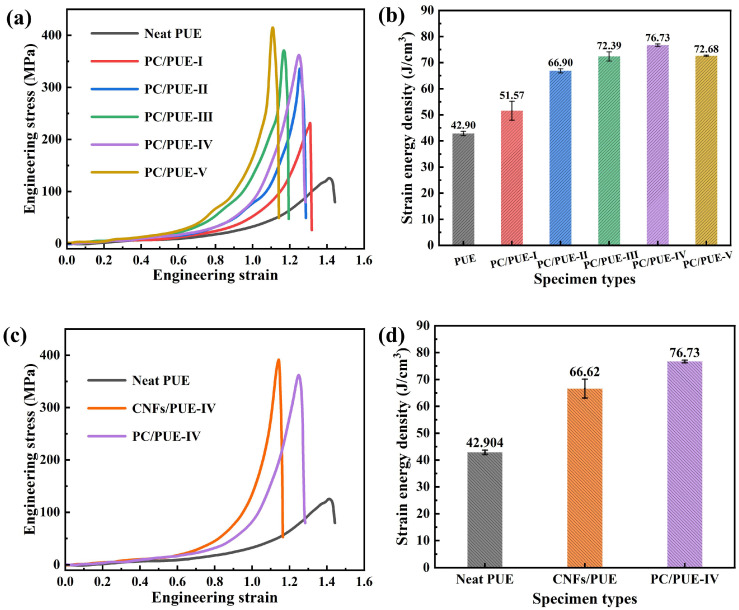
(**a**) Stress–strain curve of PUE composites in SHPB test. (**b**) Strain energy density value of dynamic impact tests of PUE composites, (**c**) Dynamic impact test stress–strain curve of PUE, 1.0% wt.% CNFs/PUE and PC/PUE–Ⅳ, (**d**) Strain energy density value of dynamic impact tests of neat PUE, CNFs/PUE and PC/PUE–Ⅳ.

**Figure 9 polymers-14-04177-f009:**
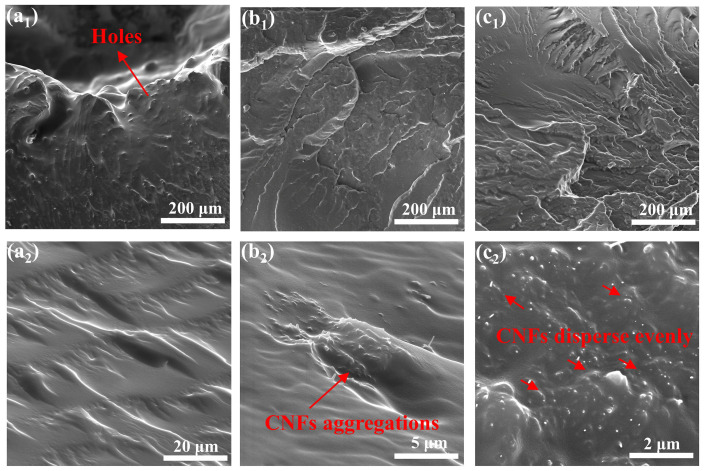
SEM images of the fracture of (**a_1_**,**a_2_**) PUE, (**b_1_**,**b_2_**) CNFs/PUE and (**c_1_**,**c_2_**) PC/PUE–Ⅳ after SHPB tests.

**Figure 10 polymers-14-04177-f010:**
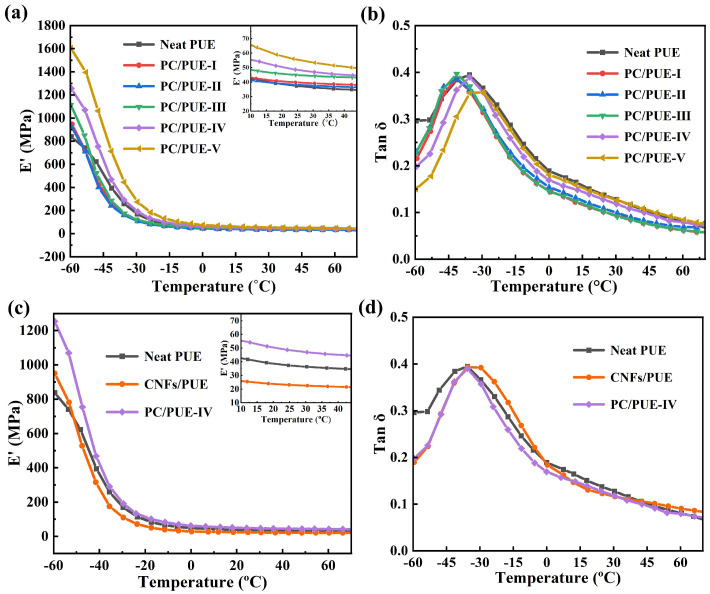
(**a**) E’ and (**b**) Tan δ as a function of temperature for Neat PUE and PDA@CNFs/PUE, (**c**) E’, and (**d**) Tan δ as a function of temperature for Neat PUE, CNFs/PUE, and PC/PUE–Ⅳ.

**Table 1 polymers-14-04177-t001:** Contents of nanofillers.

Specimens	Contents of Materials (g)		
	DA	CNFs	Tri-HCl
PDA@CNFs–1	0.20	0.70	0.10
PDA@CNFs–2	0.40	0.70	0.10
PDA@CNFs–3	0.80	0.70	0.10

**Table 2 polymers-14-04177-t002:** Contents of the PUE composites.

Specimens	Contents of Materials (g)			
	PDA@CNFs	CNFs	A Component	B Component
Neat PUE	–	–	10.00	4.00
CNFs/PUE	–	0.14	10.00	4.00
PDA@CNFs/PUE–1	0.140	–	10.00	4.00
PDA@CNFs/PUE–2	0.140	–	10.00	4.00
PDA@CNFs/PUE–3	0.140	–	10.00	4.00
PC/PUE–Ⅰ	0.028	–	10.00	4.00
PC/PUE–Ⅱ	0.070	–	10.00	4.00
PC/PUE–Ⅲ	0.112	–	10.00	4.00
PC/PUE–Ⅳ	0.140	–	10.00	4.00
PC/PUE–Ⅴ	0.168	–	10.00	4.00

**Table 3 polymers-14-04177-t003:** Data summary of PUE composites under static compression experiment.

Specimens	Compression Modulus (MPa)	Maximum Strength (MPa)	Compression Resilience Rate (%)
Neat PUE	21.21 ± 1.18	40.57 ± 2.54	45.80 ± 5.64
CNFs/PUE	26.91 ± 4.79	79.65 ± 5.66	37.14 ± 2.52
PDA@CNFs/PUE–1	43.17 ± 3.46	90.91 ± 5.26	80.90 ± 3.17
PDA@CNFs/PUE–2	34.01 ± 0.76	98.56 ± 2.94	71.43 ± 6.77
PDA@CNFs/PUE–3	24.96 ± 2.10	46.62 ± 6.73	74.30 ± 5.23
PC/PUE–Ⅰ	30.12 ± 2.13	74.12 ± 3.23	82.60 ± 5.22
PC/PUE–Ⅱ	33.12 ± 0.38	62.04 ± 5.23	71.40 ± 1.22
PC/PUE–Ⅲ	33.67 ± 1.25	68.48 ± 4.21	80.10 ± 0.81
PC/PUE–Ⅳ	43.17 ± 3.46	90.91 ± 2.51	80.90 ± 3.12
PC/PUE–Ⅴ	32.11 ± 0.25	80.94 ± 3.42	60.00 ± 0.56

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
