# Peer review of "Study on Mechanical Properties and High-Speed Impact Resistance of Carbon Nanofibers/Polyurethane Composites Modified by Polydopamine"

_polymers, 2022, doi:10.3390/polym14194177_

Round 1

Reviewer 1 Report

This paper presents some interesting experimental results about the mechanical behaviour of carbon nanofibers/polyurethane composites modify by polydopamine (PDA). However, some modifications must be performed before the work can be accepted for publication.

1)      In my opinion, the title must be changed. One think is that the idea of adding PDA can be based on mussel adhesive effects, but the authors are using, as in “the past ten years” (line 72) PDA to modify the polymers reinforcing fillers. So, I think that the title must reflect this: the use of PDA

2)      Also, in the title it is indicated that high-speed impact resistance is studied. But in the text, this is only a part, also static behaviour is considered.

3)      The nomenclature of the composites is very confusing. In point 2.3 PC/PUE-I seems to be PDA@CNFs-1 (0,2% DA) with 0,2% mass fraction of fillers. Then, with a 1% mass fraction is PC/PUE-IV. But this is also named CNFs/PUE (line 110). Then,  in point 3.2.1 appears PDA@CNFs/PUE-1 that I do not know if is the same as PC/PUE-I. I understand that the influence of DA concentration and mass rate are investigated, but I am not able to follow the nomenclature. My suggestion is to add a table with the names, the DA concentration and the mass fraction of fillers.

4)      In figure 8 there is a mistake with a “new” PC/PUE-VI

5)      Regarding figure 5: How are computed form the stress-strain curves the compression modulus and the compression resilience rate? I imagin that the compression modulus is the slope of the curve, but in which point because is a nonlinear evolution. And the resilience rate? This must be explained in the text.

6)      Figure 1 is not cited in the text

7)      Figure 3 is cited before Figure 2

8)      It is impossible to see the labels of the “experimental graphs” of figure 5 even making zoom in the pdf

Reviewer 2 Report

1-      The results have been very well presented and discussed. They are informative and interesting for readers in the field of polymer composites.

2-      Due to a proper number of tests and a large number of results, tables are really necessary for better understanding. For instance, a table for listing the samples and their content, a table for compression and impact test results, and even a final table giving all this information together.

3-      The last paragraph of the manuscript needs revision. It should briefly present the imperfections in previous literature and the remarkable distinction between your work and the innovations you offer.

4-      More reasoning is needed for explaining more details on the diagrams. The trend should be discussed in the text.

5-      A thorough spell and English check is recommended.

Round 2

Reviewer 1 Report

The authors have answered correctly of my questions. I think that the paper can be published